# Effects of Surface Rearrangement on H and O Adsorption on Cu and Pd Nanoparticles

**DOI:** 10.3390/ma18215047

**Published:** 2025-11-05

**Authors:** Nadezhda Vladimirovna Dokhlikova, Andrey Konstantinovich Gatin, Sergey Yurievich Sarvadiy, Ekaterina Igorevna Rudenko, Dinara Tastaibek, Polina Konstantinovna Ignat’eva, Maxim Vyacheslavovich Grishin

**Affiliations:** 1N.N. Semenov Federal Research Center for Chemical Physics of Russian Academy of Sciences (FRCCP RAS), Kosygina Street 4, 119991 Moscow, Russia; akgatin@yandex.ru (A.K.G.); rectedo@gmail.com (E.I.R.); polina_ignateva_2002@bk.ru (P.K.I.); mvgrishin68@yandex.ru (M.V.G.); 2Institute of Cybernetics and Information Technology, Satbaev University (KazNRTU), Satbaeva Street 22A, Almaty 050013, Kazakhstan; baimukhambetovadinara@gmail.com

**Keywords:** DFT, quantum chemical modeling, nanoparticles, transition metals, adsorption, hydrogen, oxygen

## Abstract

Atomic effects determining the adsorption of hydrogen and oxygen atoms on (111), (100), (110), and (211) surfaces of Cu and Pd have been studied using quantum chemical simulations. The deformation of the (111) and (100) surfaces during adatom bonding enhances the bond strength at active sites with high coordination numbers. In contrast, the deformation of the (110) and (211) surfaces does not exhibit a strong tendency. The atomic contribution of the nearest-neighbor environment depends on the square magnitude of the interaction matrix element, $V_{ad}^{2}$. A high $V_{ad}^{2}$ value increases the proportion of repulsive interactions within the metal adsorption complexes, leading to a decrease in the coordination number of the most stable adsorption site.

## 1. Introduction

For the past 20–30 years, nanoparticles composed of late transition and noble metals have been extensively studied by researchers across diverse fields of physics and chemistry [1,2]. This intense interest stems from both their high reactivity in a broad spectrum of reactions and the remarkable tunability of their chemical activity [3,4]. These properties render such nanoparticles a valuable platform for a wide range of functional nanomaterials [5]. While nanoparticle elemental composition can vary, copper- and palladium-based nanoobjects are of particular interest. Copper nanoparticles are currently under active investigation, driven by their intriguing physicochemical properties in nanostructured form, coupled with their low cost [6,7,8]. Palladium nanoparticles, on the other hand, have long enjoyed widespread use in the chemical industry due to their exceptional chemical activity and unique characteristics [9,10,11].

Despite extensive research on copper and palladium nanoparticles, as well as those of other transition and noble metals, using a variety of methods, the system of nanoobjects deposited on a substrate, even in a model configuration, presents a complex, multi-component environment with a highly heterogeneous surface due to the presence of interacting adsorbates [12,13]. During surface reactions, the initial adsorbates undergo dissociation, diffusion, and other transformations, occurring both through interactions with the surface of the nanoobjects and the substrate, and during the reactions themselves between surface-bound reagents [14,15]. Consequently, studying the adsorption properties of transition and noble metal nanoparticles deposited on a substrate is a highly non-trivial task, as the macroscopic parameters under investigation are influenced by a multitude of interconnected factors. Disentangling the influence of these factors requires an integrated approach. Experimentally, it is crucial to investigate changes in the electronic and atomic structure of a well-characterized nanostructured coating during the adsorption of a precisely controlled amount of gas adsorbate at the atomic level. Scanning tunneling microscopy and spectroscopy (STM/STS) offer the necessary resolution for such investigations [16,17,18]. Given that the nanoparticles used in the experiment are model nanostructures interacting with simple gas adsorbates, the STM/STS experiment is logically complemented by ab initio quantum chemical simulations. These simulations allow for the calculation of elementary surface reaction parameters and the elucidation of changes in the electronic and atomic structure of the surface as these reactions occur [19,20].

The electronic structure of transition and noble metals fundamentally governs the adsorption properties of their nanoparticles. The link between adsorption behavior and electronic structure in metal-based systems has been the subject of extensive research [21,22]. However, these studies have typically focused on either low-index surfaces or small clusters, primarily due to limitations in ab initio methodologies. Furthermore, the interpretation of such studies is often complicated by the intricate interplay between the electronic and atomic structures of nanoparticle models. Clearly, a comparative study of the adsorption properties of nanoobjects based on different metals requires the use of uniform models. Nevertheless, even with uniform models, the atomic deformation that occurs upon bond formation with an adsorbate can vary significantly depending on the metal.

The purpose of this study is to evaluate the atomic deformation contribution to the adsorption energy of hydrogen and oxygen on copper and palladium. To achieve this, the following objectives were defined:(i)Calculate the adsorption energy of hydrogen and oxygen atoms on highly symmetric active sites of flat and stepped (kinked) copper and palladium surfaces.(ii)Calculate the adsorption energy of hydrogen and oxygen atoms both with and without allowing relaxation of the surface atoms.(iii)Analyze the obtained data and quantify the contribution of atomic deformation to the adsorption energy.

Our study [23] already conducted a similar investigation into the influence of atomic deformation on adsorption energy for gold, platinum, and nickel nanoparticles. The present work serves as a comprehensive extension and continuation of this research, systematically exploring how atomic-scale factors govern the adsorption properties of nanostructures based on both transition and noble metals.

## 2. Calculation Method

Nanosystems simulating Cu and Pd nanoparticles were modeled using ab initio density functional theory (DFT) calculations with the Quantum Espresso (QE) software 7.4 package [24] within the generalized gradient approximation. The electronic structure was calculated using the PBEsol [25] approximation with the Perdew-Burke-Ernzerhof parameterization within the ultrasoft Vanderbilt pseudopotential approach [26], employing a cutoff energy for wave functions of 40 Ry and for electron density of 450 Ry [27]. All cutoff energy values were selected based on the recommendations in the corresponding pseudopotential documentation. The PBESol pseudopotential was chosen due to its more accurate reproduction of solid-state crystal lattice parameters, which is particularly relevant for this study given the use of slab models. Van der Waals interactions were accounted for using the D3 correction [28]. To integrate over the Brillouin zone, the Monkhorst-Pack method of special points [29] with an electron smearing of 0.01 Ry by the Methfessel-Paxton method [30] was selected.

Spin polarization was included in the calculation of electronic states for all systems. The total magnetization of each system was set to zero. Zero-point energy and temperature effects were not incorporated, as the analysis focused specifically on relative differences in adsorption energies between distinct active sites and the resulting trends, where absolute energy values are less critical.

Geometry optimization was performed using the Broyden–Fletcher–Goldfarb–Shanno (BFGS) algorithm. The energy convergence threshold was set to 10^−6^ Ry in the self-consistent field cycle and 10^−5^ Ry in the BFGS cycle, with a force convergence threshold of 10^−4^ Ry/bohr. The equilibrium geometry of all systems was determined at the Γ-point. A 5 × 5 × 1 *k*-point mesh was employed for the projected density of states (PDOS) calculations.

Slabs of Cu (111), (100), (110), and (211) and Pd (111), (100), (110), and (211), each consisting of 3 atomic layers and separated by a vacuum gap of approximately 10 Å under periodic boundary conditions, were used to model the surfaces of metal nanoparticles. The (111) and (100) surface models contained 48 metal atoms per supercell, the (110) models contained 64 atoms, and the (211) models contained 54 atoms. The surface areas varied depending on the metal and orientation but were on the order of 10 × 10 Å^2^, which is sufficient for calculating the adsorption of single atoms. These models represent a combination of low-index flat surfaces with varying symmetries, (111) and (100), and high-index stepped surfaces, (110) and (211), thereby mimicking the heterogeneous nature of a real nanoparticle surface.

Thus, the supercells for all surfaces possessed sufficient area to model adsorption at all high-symmetry active sites without spurious interactions with periodic images of the adsorbate. Since adsorption-induced deformation primarily occurs in the top atomic layer interacting directly with the adsorbate, two underlying layers provide adequate thickness. The vacuum gap, with a thickness twice that of the slab itself, ensures that interactions with periodic images can be neglected.

During the calculation of the atomic structure for clean slabs without adsorbates, the lattice vectors of all systems were optimized alongside the atomic positions within each monolayer. For adsorption energy calculations, computations were performed both with constrained metal atom positions and with full relaxation, where metal atoms were allowed to move freely during adsorbate optimization.

Atomic systems of slabs with hydrogen and oxygen adsorbates were calculated independently. All systems contained only a single adsorbate placed at various high-symmetry sites relative to the surface atoms, corresponding to different active sites.

The adsorbate-surface interaction is quantitatively characterized by the binding energy. In this work, unless otherwise specified, “binding” refers to chemisorption, defined as the formation of shared orbitals between the electronic shells of the adsorbate and surface metal atom(s).

The deformation of the local atomic environment during adatom adsorption occurs due to perturbation of the surface’s local electronic structure by the adsorbate. This perturbation displaces metal atoms from their initial total energy minimum configuration.

The extent of such deformation is inherently challenging to predict, as it depends on numerous interrelated parameters including the adsorbate’s position relative to metal atoms, surface orientation, packing geometry, among others.

The methodology for calculating adsorption properties on these surfaces involved determining the adsorption energy of atomic hydrogen and oxygen on highly symmetric active sites across the entire set of surfaces. These calculations were performed both with fixed metal atom positions, denoted as $E_{bond}^{f}\left(CN\right)$, and with relaxation of the metal atoms, denoted as $E_{bond}^{r}\left(CN\right)$ [31]. *CN* stands for coordination number, which is the number of nearest atoms corresponding to a given active site.(1)$$E_{bond}^{f}\left(CN\right)=E_{ads}^{f}\left(CN\right)-E_{surf}-E_{a},$$(2)$$E_{bond}^{r}\left(CN\right)=E_{ads}^{r}\left(CN\right)-E_{surf}-E_{a}.$$

Here $E_{ads}^{f}\left(CN\right)$ is the total energy of the surface with fixed metal atoms and the adsorbate, $E_{ads}^{r}\left(CN\right)$ is the total energy of the surface with relaxed (non-fixed) metal atoms and the adsorbate, $E_{surf}$ is the total energy of the clean surface, and $E_{a}$ is the total energy of the isolated adsorbate, either hydrogen or oxygen.

Calculating the binding energy at active sites with different coordination numbers without surface relaxation allows us to determine how the geometric structure of the local environment influences the adsorption energy magnitude and identify the most stable active site. By comparing these values with binding energies obtained from systems where metal atoms were fully relaxed during adsorbate addition, we can quantify the contribution of surface atomic deformation to the formation of the adsorption complex.

## 3. Results and Discussion

### 3.1. The Relationship Between Electronic and Atomic Structures

Figure 1 shows the relationship between the bond energy (adsorption energy) of H and O atoms at the T site (i.e., a one-center active site) and the position of the *d*-band center of the nearest surface metal atom. According to Norskov’s work [31], the *d*-band center for the closest-packed surface of copper is −2.67 eV, and for palladium it is −1.83 eV. The binding energy for an oxygen atom monolayer with copper is −4.5 eV, and with palladium it is −4.2 eV [31]. As can be seen, the relative differences between the metals align with the values obtained in the present study. The observed discrepancies are clearly associated with differences in the construction of the surface and adsorbate models.

When comparing the *d*-band centers and adsorption energies of hydrogen atoms on surfaces of different metals with similar atomic packing, a higher adsorption energy corresponds to a *d*-band center closer to the Fermi level. Thus, hydrogen atoms bind more strongly to palladium surfaces than to copper surfaces with similar atomic packing. However, oxygen adsorption exhibits the opposite trend across all copper and palladium surfaces.

The adsorption behavior of these metals depends not only on the adsorbent but also on the adsorbate. All subsequent theoretical analysis and mathematical formulations are based on the established framework developed by Norskov and colleagues, as referenced in [31,32]. In a simplified, one-electron approximation, the total bond energy of an adatom to the surface of a transition metal can be expressed as the sum of two contributions [32]:(3)$$E_{bond}=E_{s}+E_{d},$$
where $E_{s}$ is the interaction of the adsorbate with the metal’s *s*-band, $E_{d}$ is the interaction of the adsorbate with the metal’s *d*-band. According to Nørskov’s theory of resonant chemisorption [32], the first contribution, $E_{s}$, is associated with ‘weak chemisorption,’ leading to a downward shift in energy and a broadening of the adsorbate level. Given that all transition metals possess a single electron in their *s*-orbital, this contribution can be considered relatively constant across the transition metal series. The second contribution, $E_{d}$, is associated with ‘strong chemisorption,’ involving the formation of bonding and antibonding orbitals at the lower and upper edges of the *d*-band. This contribution is sensitive to the filling of the *d*-band and dictates the variations in adsorption behavior among transition metals. Consequently, due to the presence of both bonding and antibonding orbitals, the interaction with the *d*-band represents a balance between attractive and repulsive forces.(4)$$E_{d}=E_{att}+E_{rep}.$$

For the interaction of a single adsorbate level with a *d*-band of filling *f*, the contribution from the *d*-band can be expressed as:(5)$$E_{d}=-2\left(1-f\right)\frac{\beta V_{ad}^{2}}{\left|\varepsilon_{d}-\varepsilon_{ads}\right|}+2\left(1+f\right)\alpha \beta V_{ad}^{2},$$
where the first term accounts for attraction and the second term accounts for repulsion. The coefficient *β*, which depends on the adsorbate, is included in both terms. The parameter *α*, representing the overlap of metal orbitals, exhibits only slight variation across transition metals and is included in the repulsion term. The matrix coupling element, $V_{ad}$, describes the overlap between the orbitals of the adsorbate and the metal and is included in both terms.

Copper has a *d*-band filling of *f* = 1; therefore, the *d*-band contribution upon adsorption of any adsorbate will primarily consist of repulsion.(6)$$E_{bond}\left(Cu\right)=E_{s}+E_{rep}\left(Cu\right),$$(7)$$E_{rep}\left(Cu\right)=4\alpha \beta V_{ad}^{2}\left(Cu\right).$$

For palladium, the *d*-band filling is *f* = 0.9, resulting in an adsorption energy with both attractive and repulsive contributions.(8)$$E_{bond}\left(Pd\right)=E_{s}+E_{att}\left(Pd\right)+E_{rep}\left(Pd\right),$$(9)$$E_{att}\left(Pd\right)=-0.2\frac{\beta V_{ad}^{2}}{\left|\varepsilon_{d}\left(Pd\right)-\varepsilon_{ads}\right|}\;,$$(10)$$E_{rep}(Pd)=3.8\alpha \beta V_{ad}^{2}(Pd).$$

Given that the $E_{s}$ contribution is approximately the same for copper and palladium, the differences in adsorption energies are primarily determined by the $E_{d}$ contribution. Although copper lacks an attractive contribution, its interaction matrix element of is relatively small ($V_{ad}$ = 1) [31], resulting in a modest repulsive force, even when considering adsorbates with more extended orbitals, such as oxygen. In contrast, palladium exhibits a considerably larger interaction matrix element of ($V_{ad}$ = 2.78) [31], which, coupled with a high *d*-band filling (*f* = 0.9), creates a competitive environment where the adsorbate’s characteristics, represented by its *β* value, play a significant role. This explains the observed differences in the trends of adsorption energy for hydrogen and oxygen.

Furthermore, considering surfaces with different atomic arrangements of the same metal reveals subtle deviations from the trend described above. This is because the active sites on these surfaces possess varying symmetries and distinct immediate environments. Moreover, the stepped surfaces (110) and (211) feature atoms with inequivalent nearest-neighbor environments. For instance, on the (211) surface, an adsorbate atom can reside at the edge, middle, or in the depth of a step, forming a bond with a different number of neighboring atoms.

Thus, while the electronic factor associated with the *d*-band filling and the position of its center undeniably plays a dominant role in determining the adsorption energy of adatoms on the surface, the influence of atomic factors, particularly when considering deformed and stepped surfaces of a single metal, cannot be neglected. Here, the atomic factors encompass both the immediate environment of the active site and the surface deformation induced by adatom adsorption. The influence of the atomic environment is investigated by calculating the adsorption energies at active sites with different coordination numbers while maintaining fixed positions for the metal atoms on the surface. Incorporating the optimization of atom positions on the surface during adsorption at the same active sites accounts for the atomic deformation of the surface.

Flat (111) and (100) surfaces, and kinked (110) and (211) surfaces, exhibit different sets of highly symmetric active sites, as illustrated in Figure 2. The (111) surface features one-center T, two-center B, and three-center HCP and FCC active sites. The (100) surface features one-center T, two-center B, and four-center H active sites. Kinked surfaces possess a more diverse structure. The (110) surface presents two one-center sites (T1, T2), two distinct types of two-center sites (Bl1, Bl2 and Bh1, Bh2) with varying bond lengths, and two three-center sites (H1, H2). The (211) surface boasts an even greater variety of active sites due to the step structure, including one-center sites (T1, T2, T3), two-center sites (Bl1, Bl2, Bl3, Bl4 and Bh1, Bh2, Bh3 and Br), three-center sites (H1, H2, H3, H4, Hr), and a four-center site (R).

### 3.2. Atomic Contributions of Highly Symmetric Hydrogen and Oxygen Adsorption Sites on Cu and Pd(111), (100), (110), and (211) Surfaces

#### 3.2.1. Cu(111), Cu(100), Cu(110), Cu(211)

Figure 3 illustrates energy maps, adsorption energies, and energy profiles for H and O atoms on Cu(111), Cu(100), Cu(110), and Cu(211) surfaces. A significant difference in the adsorption energies of H and O atoms is evident, ranging from approximately 1.5 to 4 eV (Table 1). Oxygen forms a stronger bond with any Cu surface compared to hydrogen. Here and throughout this work, the term “diffusion” refers not to the actual diffusion process, but to the lower bound estimate of its energy barrier—specifically, the energy difference between distinct active sites. Hydrogen diffusion barriers for all surfaces range from 0.4 to 0.8 eV, which is less than 1 eV. Conversely, for oxygen, these barriers can exceed 2 eV. Notably, on the (110) surface, the oxygen diffusion barrier does not exceed 0.5 eV, similar to the hydrogen diffusion barrier. Low diffusion barriers and high chemical activity for the Au(110) surface have been reported in [33]. However, for copper surfaces, the adsorption energies of hydrogen and oxygen themselves exhibit only slight variations between different surface orientations.

Figure 4 illustrates the change in the projected density of states (PDOS) of a Cu (111) surface atom upon H and O adsorption. The graph reveals a filled *d*-band and the emergence of bonding and antibonding states at the upper and lower edges of the *d*-band following interaction with the adatoms. As is evident, the PDOS shifts are similar for both adatoms, indicating that the higher adsorption energy of oxygen is primarily attributable to its stronger interaction with the copper *s*-band. This is particularly pronounced due to the low value of the copper-adsorbate coupling matrix element, *V_ad_* compared to other noble metals.

The contribution of atomic relaxation to the adsorption energy of adsorbates on surfaces exhibits different trends for flat (111), (100) and kinked (110), (211) Cu surfaces. As shown in Figure 3a–d, the atomic contributions from the immediate environment and surface deformation to the adsorption energy of the H, HCP, and FCC active sites are greater than those of the B and T active sites. In other words, the atomic contributions increase with increasing coordination number of the active site for flat (111) and (100) Cu surfaces. This trend is anticipated, as the optimal position of the adsorbate on Cu is at the maximum distance from the surface atoms, which is achieved during adsorption at two- and three-center active sites. This suggests a limited overlap of the orbitals of the adsorbate and adsorbent and, consequently, a small contribution of repulsive interactions to bond formation.

For kinked (110) and (211) surfaces, a direct correlation between the atomic contribution from the immediate environment and the coordination number is not observed. In general, the atomic contribution from surface relaxation is noticeably smaller than that for flat surfaces, not exceeding 0.2–0.4 eV for hydrogen and approximately 1 eV for oxygen. This finding aligns with data obtained for nickel surfaces published in our previous work [29]. Nickel also possesses a small $V_{ad}$ value, but as a transition metal with an unfilled *d*-band, it benefits from an attractive contribution from the *d*-shell, resulting in a stronger interaction with the adsorbate. This was evident in the interaction with uneven surfaces, manifesting not only as higher adsorption energies but also as a greater atomic contribution to the overall adsorption energy. It can be hypothesized that the attractive contribution from the *d*-shell, rather than the *s*-shell, plays a more significant role in the synergy between atomic and electronic factors in bond formation, as highlighted in previous work.

#### 3.2.2. Pd(111), Pd(100), Pd(110), Pd(211)

Figure 5 illustrates the energy maps, adsorption energies, and energy profiles of H and O atoms on Pd(111), Pd(100), Pd(110), and Pd(211) surfaces. The adsorption energies of H and O on Pd surfaces differ by approximately 3 eV for both flat and kinked surfaces. While the bonding of O to Pd is weaker than the bonding of O to Cu, and the bonding of H to Pd is stronger than the bonding of H to Cu, the underlying reasons for these differences have been discussed previously. Hydrogen diffusion barriers on all surfaces do not exceed 0.5 eV (Table 2). It is well known that palladium can accumulate hydrogen in its bulk, which might suggest even lower diffusion barriers. However, surface and subsurface diffusion are similar yet distinct phenomena, as the electronic structure of the surface differs from that of the bulk. Diffusion into the bulk was not investigated in this work. Notably, oxygen diffusion barriers on kinked surfaces exhibit similar values, remaining below 0.5 eV. For flat surfaces, this value increases to approximately 1 eV. Palladium exhibits more facile diffusion over kinked surfaces.

Figure 6 illustrates the change in the projected density of states (PDOS) of the Pd(111) surface atom upon H and O adsorption. A larger shift in the density of *d*-states during O adsorption correlates with a greater adsorption energy. The *d*-band of Pd exhibits a broader density distribution compared to Cu, indicating a larger $V_{ad}^{2}$ value [21,22]. This is because the value of the interaction matrix element is proportional to the bandwidth.

Distinct trends in atomic contributions to the adsorption energy are observed on flat (111), (100) and kinked (110), (211) surfaces. For the flat Pd(111) and Pd(100) surfaces, the atomic contribution from surface relaxation is proportional to the coordination number of the active site, similar to the Cu surface. Furthermore, in the absence of surface relaxation (Figure 5a–d), one-center active sites are found to be more stable. The adsorbate tends to occupy a position closer to the adsorbent atoms.

However, in immediate environments containing multiple Pd atoms, the repulsive interaction increases with increasing coordination number. This makes the one-center T site more favorable, despite the shorter bond length, but only on an undeformed surface. Surface relaxation during adsorption reverses this trend, as in the case of flat copper surfaces, where it is more advantageous for the adsorbate to form a bond with a larger number of metal atoms. Evidently, this behavior arises from the higher $V_{ad}^{2}$ value of palladium compared to copper.

On the highly kinked (211) surface, no overarching trend can be discerned. The atomic contribution increases the adsorption energy by approximately 0.8–1.0 eV for both adsorbates, which is comparable to the atomic contribution observed on flat (111) and (100) surfaces. However, the (110) surface exhibits a more unusual characteristic in terms of diffusion, specifically at the Bh1, Bh2 and H1 active sites, where the atomic contribution approaches nearly 2–3 eV. Similar behavior for this active site has been previously reported for Pt [29]. In general, diffusion on stepped Pd surfaces shares similarities with diffusion on platinum. This suggests that palladium surfaces exhibit a synergy between electronic and atomic structures, although the increase in the atomic contribution for the (110) and (211) surfaces is not as pronounced as it is for platinum.

## 4. Conclusions

Based on the conducted study of the adsorption properties of (111), (100), (110), and (211) surfaces through the interaction with hydrogen and oxygen atoms on Cu and Pd, the following conclusions can be drawn:(i)The influence of atomic deformation on adsorption energy is most pronounced for active sites with *CN* = 3 on flat (111) surfaces and *CN* = 4 on (100) surfaces. For curved (110) and (211) surfaces, the effect of atomic deformation shows no clear trend. This behavior appears to be governed by surface packing density. Surfaces with higher packing density—and consequently more uniform local environments around each active site—exhibit maximum deformation effects at sites with the highest coordination number. In contrast, surfaces with lower packing density contain active sites with heterogeneous local coordination environments, where the influence of atomic deformation varies even among sites with identical coordination numbers but different atomic arrangements.(ii)Atomic deformation consistently enhances binding strength across all studied systems.(iii)The influence of the local atomic environment depends on the adatom type and indirectly correlates with the magnitude of the interaction matrix element. A larger interaction matrix element corresponds to a greater contribution of repulsive interactions in the metal-adsorbate complex. When repulsive interactions have limited influence, the contribution from surrounding atoms is proportional to the active site’s coordination number. Conversely, strong repulsive interactions reverse this dependence. A similar correlation between the local environment effect and the interaction matrix element was also established for gold, platinum, and nickel.(iv)Our results demonstrate that atomic-scale contributions can play a significant role in determining the reactivity and adsorption properties of nanoscale systems. Consequently, when selecting catalytic systems, not only the chemical composition of the nanophase but also the choice of its support material becomes crucial. This insight opens new possibilities for enhancing and fine-tuning the properties of practical catalytic nanosystems.

## Figures and Tables

**Figure 1 materials-18-05047-f001:**
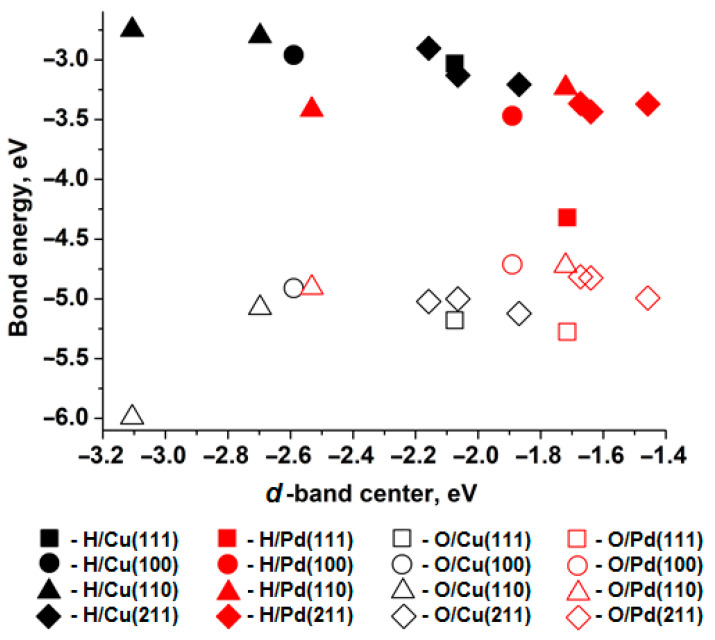
Relationship between the bond energy of H adatoms (solid figures) and O (empty shapes) position of the *d*-band center of the nearest metal atom of surface (111) (squares), (100) (circles), (110) (triangles), (211) (diamonds), Cu—black shapes, Pd—red shapes.

**Figure 2 materials-18-05047-f002:**
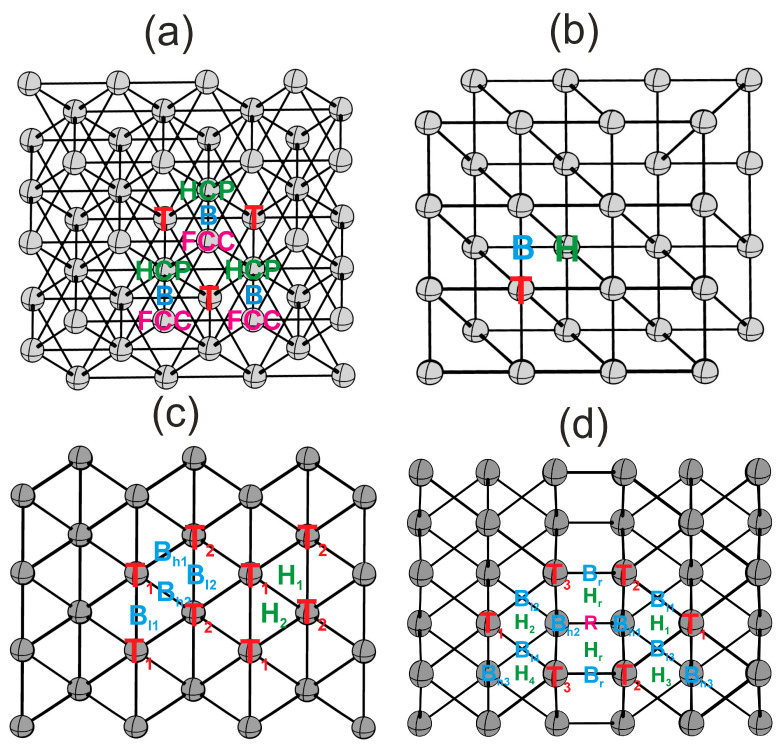
Highly symmetric active sites on surfaces: (**a**)—(111), (**b**)—(100), (**c**)—(110), (**d**)—(211). Colors indicate active site types: T—red, B—blue, H, HCP—green, FCC, R—pink [23].

**Figure 3 materials-18-05047-f003:**
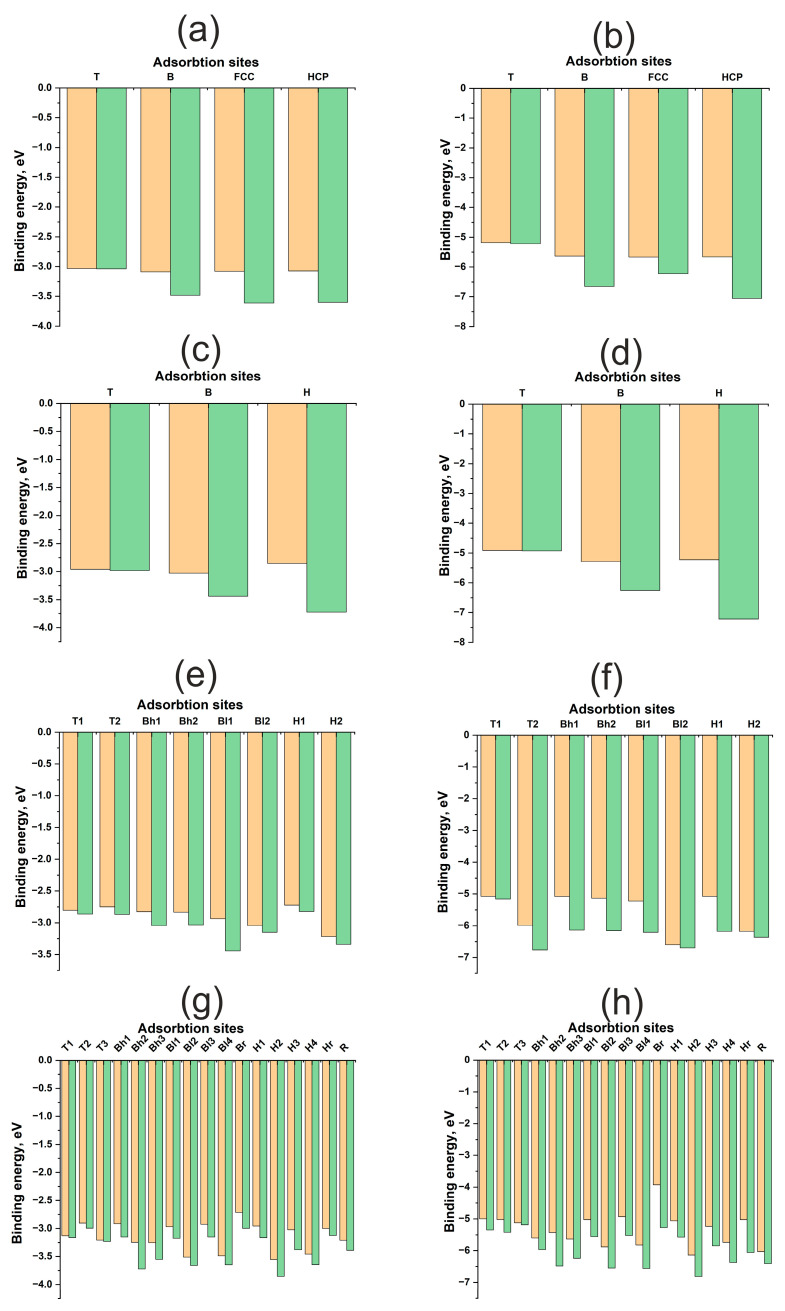
Adsorption energies for different active sites for H and O atom on Cu surfaces: (**a**)—H on Cu (111), (**b**)—O on Cu (111), (**c**)—H on Cu (100), (**d**)—O on Cu (100), (**e**)—H on Cu (110), (**f**)—O on Cu (110), (**g**)—H on Cu (211), (**h**)—O on Cu (211). The orange columns represent $E_{bond}^{f}$, and the green columns represent $E_{bond}^{r}$.

**Figure 4 materials-18-05047-f004:**
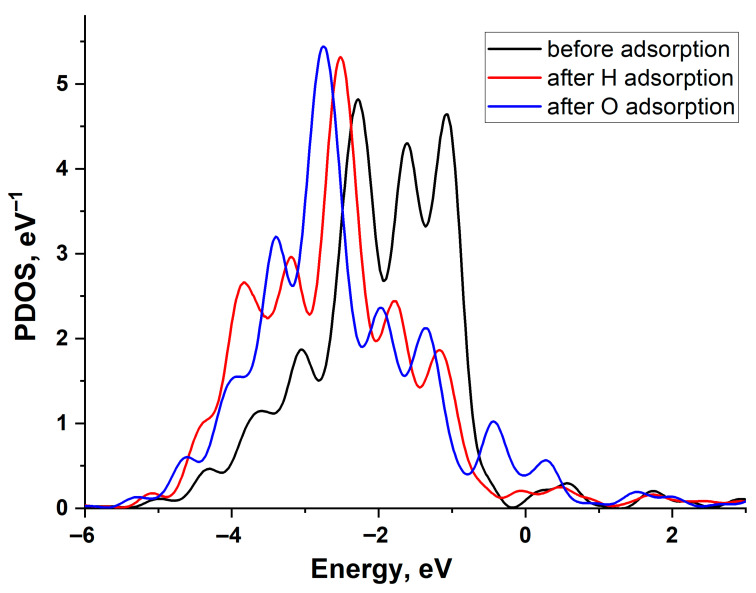
Projected density of states of the Cu(111) surface atom: black line—before adsorption, red line—after H adsorption, and blue line—after O adsorption.

**Figure 5 materials-18-05047-f005:**
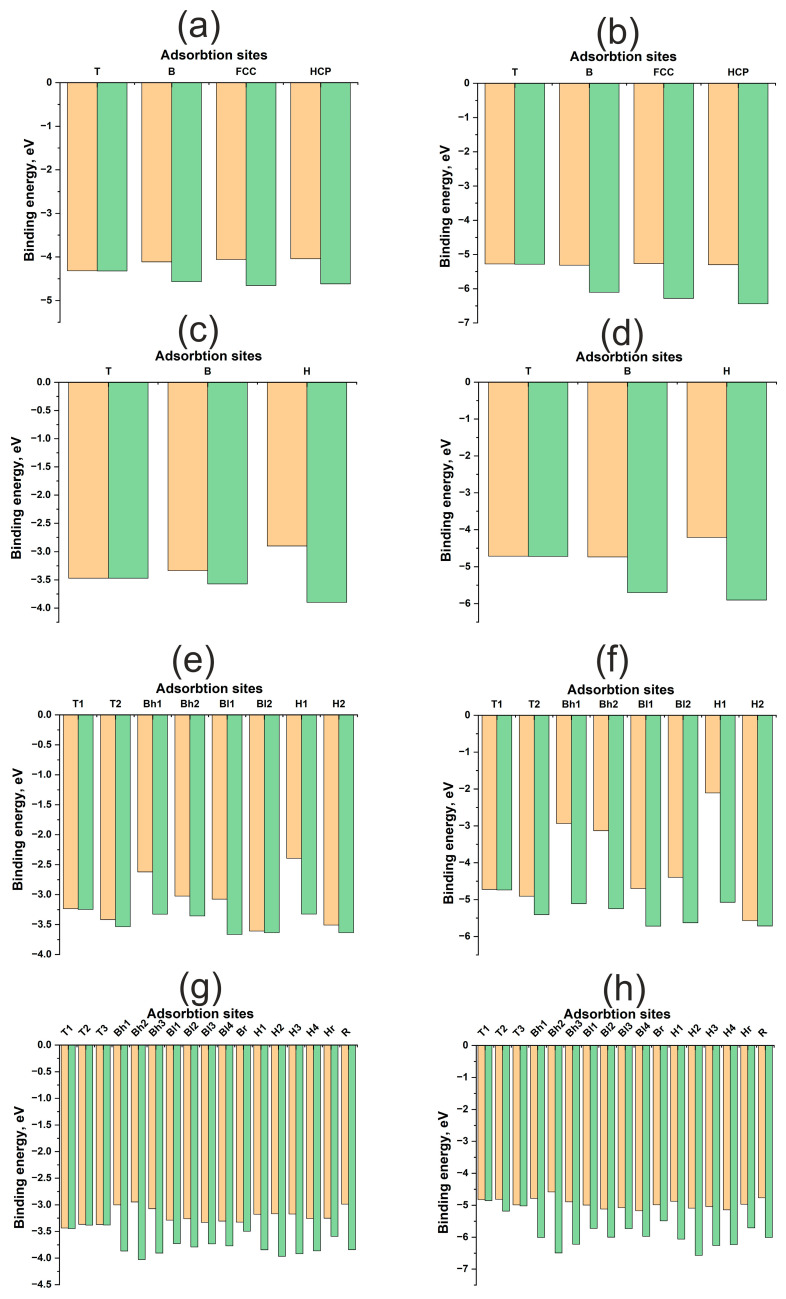
Adsorption energies for various active sites for H and O atom on Pd surfaces: (**a**)—H on Pd (111), (**b**)—O on Pd (111), (**c**)—H on Pd (100), (**d**)—O on Pd (100), (**e**)—H on Pd (110), (**f**)—O on Pd (110), (**g**)—H on Pd (211), (**h**)—O on Pd (211). The orange columns represent $E_{bond}^{f}$, and the green columns represent $E_{bond}^{r}$.

**Figure 6 materials-18-05047-f006:**
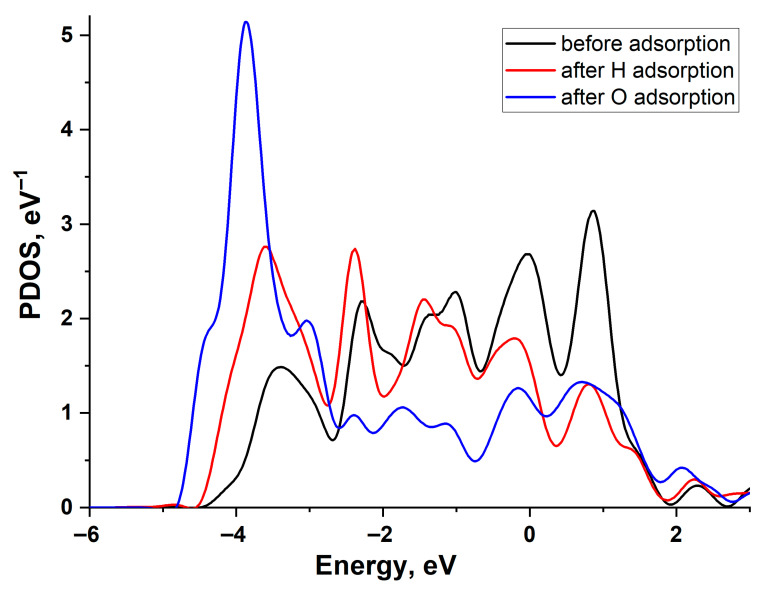
Projected density of states of the Pd(111) surface atom: black line—before adsorption, red line—after H adsorption, and blue line—after O adsorption.

**Table 1 materials-18-05047-t001:** Bond energies $E_{bond}^{r}$ (eV), bonding distances $R_{bond}^{r}$ (Å), distances to the metal surfaces $R_{surf}^{r}$ (Å), of H and O adatoms for various active sites of Cu(111), Cu(100), Cu(110), Cu(211) surfaces.

	H				O			
Cu(111)	$E_{bond}^{f}$	$E_{bond}^{r}$	$R_{bond}^{r}$	$R_{surf}^{r}$	$E_{bond}^{f}$	$E_{bond}^{r}$	$R_{bond}^{r}$	$R_{surf}^{r}$
T	−3.03	−3.04	1.51	1.54	−5.18	−5.22	1.72	1.71
B	−3.09	−3.48	1.66	1.14	−5.64	−6.65	1.82	1.24
FCC	−3.08	−3.61	1.74	1.03	−5.67	−6.22	1.87	1.18
HCP	−3.07	−3.60	1.74	1.00	−5.66	−7.06	1.98	1.33
Cu(100)	$E_{bond}^{f}$	$E_{bond}^{r}$	$R_{bond}^{r}$	$R_{surf}^{r}$	$E_{bond}^{f}$	$E_{bond}^{r}$	$R_{bond}^{r}$	$R_{surf}^{r}$
T	−2.96	−2.98	1.521	1.609	−4.91	−4.93	1.71	1.76
B	−3.03	−3.44	1.642	1.179	−5.29	−6.26	1.80	1.36
H	−2.85	−3.73	1.832	0.599	−5.23	−7.22	1.97	0.96
Cu(110)	$E_{bond}^{f}$	$E_{bond}^{r}$	$R_{bond}^{r}$	$R_{surf}^{r}$	$E_{bond}^{f}$	$E_{bond}^{r}$	$R_{bond}^{r}$	$R_{surf}^{r}$
T1	−2.80	−2.86	1.52	1.595	−5.08	−5.16	1.70	1.67
Bl1	−2.94	−3.44	1.64	1.25	−5.22	−6.21	1.80	1.46
Bh2	−2.83	−3.04	1.99	1.58	−5.14	−6.16	1.95	1.465
Bh1	−2.83	−3.05	1.995	1.62	−5.08	−6.14	1.96	1.50
Bl2	−3.05	−3.15	1.76	1.31	−6.60	−6.70	2.06	1.65
T2	−2.75	−2.87	1.695	1.63	−5.99	−6.77	2.06	1.88
H2	−3.22	−3.34	1.93	1.46	−6.18	−6.36	1.96	1.45
H1	−2.72	−2.82	1.79	1.165	−5.08	−6.18	1.96	1.08
Cu(211)	$E_{bond}^{f}$	$E_{bond}^{r}$	$R_{bond}^{r}$	$R_{surf}^{r}$	$E_{bond}^{f}$	$E_{bond}^{r}$	$R_{bond}^{r}$	$R_{surf}^{r}$
Bh1	−2.92	−3.15	1.69	1.15	−5.60	−5.97	1.97	1.45
H1	−2.96	−3.16	1.74	0.997	−5.06	−5.57	1.91	1.30
T1	−3.13	−3.16	1.53	1.52	−5.00	−5.35	1.81	1.67
H2	−3.55	−3.85	1.75	1.06	−6.14	−6.81	1.89	1.31
Bh2	−3.24	−3.72	1.63	1.14	−5.43	−6.49	1.79	1.38
R	−3.21	−3.39	2.15	2.79	−6.03	−6.41	2.17	1.31
Bl1	−2.97	−3.17	1.67	1.13	−5.02	−5.56	1.87	1.35
Bl2	−3.51	−3.66	1.69	1.17	−5.88	−6.55	1.87	1.28
Hr	−3.00	−3.12	2.00	1.49	−5.03	−6.06	1.90	1.38
T2	−2.90	−2.99	1.59	1.55	−5.02	−5.42	1.88	1.77
H3	−3.02	−3.38	1.73	1.08	−5.23	−5.85	1.899	1.29
Bh3	−3.25	−3.55	1.69	1.17	−5.63	−6.24	1.89	1.42
H4	−3.46	−3.64	1.79	1.09	−5.73	−6.37	1.91	1.24
T3	−3.21	−3.23	1.50	1.52	−5.12	−5.19	1.69	1.73
Br	−2.71	−3.00	2.11	1.87	−3.93	−5.28	1.83	1.60
Bl3	−2.93	−3.15	1.67	1.18	−4.93	−5.52	1.87	1.35
Bl4	−3.49	−3.65	1.69	1.17	−5.83	−6.56	1.87	1.27

**Table 2 materials-18-05047-t002:** Bond energies $E_{bond}^{r}$ (eV), bonding distances $R_{bond}^{r}$ (Å), distances to the metal surfaces $R_{surf}^{r}$ (Å), of H and O adatoms for various active sites of Pd(111), Pd(100), Pd(110), Pd(211) surfaces.

	H				O			
Pd(111)	$E_{bond}^{f}$	$E_{bond}^{r}$	$R_{bond}^{r}$	$R_{surf}^{r}$	$E_{bond}^{f}$	$E_{bond}^{r}$	$R_{bond}^{r}$	$R_{surf}^{r}$
T	−4.32	−4.32	1.54	1.54	−5.28	−5.29	1.83	1.79
B	−4.11	−4.57	1.73	1.11	−5.31	−6.10	1.96	1.40
FCC	−4.04	−4.62	1.833	0.99	−5.26	−6.28	2.02	1.24
HCP	−4.06	−4.66	1.83	1.00	−5.30	−6.44	2.03	1.32
Pd(100)	$E_{bond}^{f}$	$E_{bond}^{r}$	$R_{bond}^{r}$	$R_{surf}^{r}$	$E_{bond}^{f}$	$E_{bond}^{r}$	$R_{bond}^{r}$	$R_{surf}^{r}$
T	−3.47	−3.47	1.544	1.554	−4.71	−4.72	1.805	1.802
B	−3.34	−3.89	1.709	1.083	−4.74	−5.70	1.917	1.389
H	−2.90	−3.90	1.945	0.575	−4.21	−5.90	2.114	0.991
Pd(110)	$E_{bond}^{f}$	$E_{bond}^{r}$	$R_{bond}^{r}$	$R_{surf}^{r}$	$E_{bond}^{f}$	$E_{bond}^{r}$	$R_{bond}^{r}$	$R_{surf}^{r}$
T1	−3.24	−3.25	1.537	1.475	−4.72	−4.74	1.788	1.79
Bl1	−3.08	−3.66	1.691	1.059	−4.70	−5.72	1.894	1.406
Bh2	−3.02	−3.36	2.140	1.791	−3.13	−5.25	2.401	1.96
Bh1	−2.62	−3.33	2.198	1.87	−2.94	−5.11	2.447	2.05
Bl2	−3.61	−3.64	2.027	1.52	−4.39	−5.63	2.978	2.604
T2	−3.42	−3.54	1.715	1.659	−4.91	−5.40	2.166	2.038
H2	−3.51	−3.64	1.994	1.407	−5.57	−5.72	2.256	1.882
H1	−2.39	−3.33	2.513	1.97	−2.10	−5.07	2.769	2.089
Pd(211)	$E_{bond}^{f}$	$E_{bond}^{r}$	$R_{bond}^{r}$	$R_{surf}^{r}$	$E_{bond}^{f}$	$E_{bond}^{r}$	$R_{bond}^{r}$	$R_{surf}^{r}$
Bh1	−3.00	−3.87	1.705	0.771	−4.79	−6.01	1.962	1.228
H1	−3.18	−3.85	1.799	0.809	−4.88	−6.06	1.974	1.11
T1	−3.44	−3.45	1.543	1.996	−4.82	−4.85	1.812	1.793
H2	−3.19	−3.97	1.790	0.794	−5.09	−6.57	1.975	1.128
Bh2	−2.95	−4.03	1.69	0.431	−4.59	−6.49	1.911	0.981
R	−2.99	−3.84	1.996	0.348	−4.77	−6.01	2.22	1.004
Bl1	−3.29	−3.73	1.709	1.085	−5.00	−5.72	1.921	1.325
Bl2	−3.26	−3.79	1.700	1.047	−5.12	−6.00	1.917	1.338
Hr	−3.25	−3.59	1.724	0.827	−4.97	−5.70	1.963	1.152
T2	−3.37	−3.38	1.559	1.184	−4.81	−5.19	1.881	1.793
H3	−3.17	−3.92	1.794	0.822	−5.04	−6.25	1.986	1.197
Bh3	−3.07	−3.91	1.702	0.744	−4.89	−6.22	1.95	1.124
H4	−3.28	−3.86	1.813	0.858	−5.14	−6.23	1.983	1.097
T3	−3.37	−3.38	1.546	1.989	−4.99	−5.02	1.789	1.852
Br	−3.33	−3.50	1.717	1.095	−4.99	−5.49	1.898	1.3
Bl3	−3.33	−3.73	1.708	1.069	−5.08	−5.73	1.921	1.35
Bl4	−3.31	−3.77	1.704	1.056	−5.17	−5.97	1.917	1.321

## Data Availability

The original contributions presented in this study are included in the article. Further inquiries can be directed to the corresponding author.

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
