# Peer review of "Effects of Surface Rearrangement on H and O Adsorption on Cu and Pd Nanoparticles"

_materials, 2025, doi:10.3390/ma18215047_

Round 1
Reviewer 1 Report
Comments and Suggestions for Authors
The manuscript presented to me for the review process investigates the atomic effects governing the adsorption of hydrogen and oxygen atoms on various Cu and Pd surfaces using quantum chemical simulations. The study provides valuable insights into how surface deformation and coordination environments influence adsorption strength and stability. I would like to thank the authors for their constructive and well-motivated research on this relevant topic.
The manuscript is carefully prepared and contains valuable information. The article could be published in Materials, but it should be improved beforehand. My decision: consider for publication after minor revisions. Below are several comments that may help enhance the manuscript:
- The topic is relevant, but its novelty relative to Dokhlikova et al., Materials 2025, is unclear. Please explicitly highlight how your results differ or extend previous work, especially regarding Cu and Pd adsorption trends.
- Provide more details on pseudopotentials, functional, dispersion corrections, k-points, cutoff energies, and optimization criteria. This ensures reproducibility and strengthens methodology credibility.
- Justify the slab thickness and vacuum spacing. Clarify which layers were relaxed and how convergence of adsorption energies was tested.
- Clearly define all energy terms (bond, deformation, enhancement) and their physical meaning. Specify which terms include surface relaxation.
- Explain how V_ad was determined and provide numerical examples for Cu and Pd. Support claims about attractive vs. repulsive contributions quantitatively.
- Include systematic comparisons with previously reported adsorption energies and site preferences. This validates your computational setup and observed trends.
- Discuss why high-coordination sites on some surfaces show stronger bonding while others do not. Connect findings to local coordination and d-band properties.
- Indicate if the observed Vad2_adsorption correlation is general or specific to Cu/Pd. Relate to d-band center or orbital symmetry considerations.
- Clearly differentiate new results from previously published work. Cite related studies and explicitly describe the unique contribution of this study.
- Mention whether spin polarization, zero-point energy, or temperature effects were included. Clarify if H and O adsorption configurations were optimized independently and note any coverage effects.
Reviewer 2 Report
Comments and Suggestions for Authors
See attached file.

Reviewer 3 Report
Comments and Suggestions for Authors
This manuscript presents DFT calculations to investigate the adsorption of H and O on different Cu and Pd surfaces, and surface relaxation and atomic environment contributions to adsorption energy were analyzed and clearly demonstrated. The methodology is appropriate and described in detail. Overall, I recommend accepting this manuscript after minor revisions. Here are my comments and concerns:
- It would enhance the readability of this paper if the authors could add clear legends for Figure 1, 3, 4, 5 and 6. Color-coded labels are recommended.
- I recommend adding a summary table to compare adsorption energies and diffusion barriers across all surfaces.
- In the section of different Pd surface, the authors mentioned that “palladium is unique among the transition metals”, more discussion is needed.
- In conclusion, the authors repeated earlier points, I recommend adding more discussion about potential implications for catalytic activity, stability, and surface design strategies.
- Please check the DOIs of references, reference 3 and 22 are using the same DOI.
Reviewer 4 Report
Comments and Suggestions for Authors
The manuscript by Dokhlikova et al. employs density functional theory (DFT) calculations to analyze hydrogen and oxygen adsorption on Cu and Pd surfaces, (111), (100), (110), and (211), explicitly distinguishing the effects of nearest-neighbor coordination (fixed slabs) from those of surface relaxation (relaxed slabs). The results show that surface relaxation strengthens adsorption primarily at high-coordination sites on flat (111) and (100) facets, whereas on stepped (110) and (211) surfaces, the trends become site-dependent and less systematic. This dual treatment successfully isolates electronic and atomic deformation contributions across two metals and four facets.
However, the computational scope is limited by the thin (three-layer) slabs, small supercells, and lack of convergence validation. Such thin slabs may inadequately represent the bulk electronic structure, leading to uncertain absolute adsorption energies and diffusion barriers. Improved numerical reliability could be achieved by employing thicker slabs, larger vacuum spacing and k-meshes, and higher plane-wave cutoffs or PAW potentials.
The simulations address single-atom adsorption at 0 K, neglecting finite-temperature and coverage effects that are essential for realistic catalytic conditions. Inclusion of zero-point energy (ZPE) and entropy corrections, as well as the treatment of coadsorbate interactions (e.g., O + H and OH formation), would make the thermodynamic and kinetic trends more meaningful.
Although the models are described as “nanoparticles on graphite,” the calculations actually use planar metal slabs without explicit substrate atoms. Curvature, edge sites, and charge-transfer effects from graphitic supports, which can significantly alter adsorption and relaxation behavior, are therefore not represented.
Furthermore, spin polarization and reference states for oxygen are not discussed, potentially affecting all O-related adsorption and diffusion results.
The energy profiles presented lack methodological specification, whether based on NEB, CI-NEB, or simple interpolation, raising uncertainty in the reported barrier heights, which could be path-dependent or underestimated.
Finally, the reported ~5 eV relaxation contribution for the Pd(110)-H1 site appears physically unrealistic, exceeding typical adsorption energies by several electronvolts and likely reflecting numerical or decomposition errors.
Round 2
Reviewer 4 Report
Comments and Suggestions for Authors
Required changes are made by the authors. The revision is in a better shape for publication. The revision cab be accepted.